# Study protocol for resolution of organ injury in acute pancreatitis (RESORP): an observational prospective cohort study

Ahmed E Sherif,[1] Rory McFadyen,[1] Julia Boyd,[2] Chiara Ventre,[1] Margaret Glenwright,[3] Kim Walker,[3] Xiaozhong Zheng,[4] Audrey White,[3] Laura McFadyen,[3] Emma Connon,[3] Dimitrios Damaskos,[1] Michelle Steven,[2] Anthony Wackett,[2] Euan Thomson,[5] David C Cameron,[5] Jill MacLeod,[6] Shaun Baxter,[6] Scott Semple,[7] David Morris,[4] Saskia Clark-Stewart,[1] Catriona Graham,[8] Damian J Mole ![ORCID] ,[1,4] On behalf of the RESORP research team

AES, RM and JB are joint first authors.

For numbered affiliations see end of article.

**Correspondence to**
Professor Damian J Mole;
damian.mole@ed.ac.uk

## ABSTRACT

**Introduction** Survivors of acute pancreatitis (AP) have shorter overall survival and increased incidence of new-onset cardiovascular, respiratory, liver and renal disease, diabetes mellitus and cancer compared with the general population, but the mechanisms that explain this are yet to be elucidated. Our aim is to characterise the precise nature and extent of organ dysfunction following an episode of AP.

**Methods and analysis** This is an observational prospective cohort study in a single centre comprising a University hospital with an acute and emergency receiving unit and clinical research facility. Participants will be adult patient admitted with AP. Participants will undergo assessment at recruitment, 3 months and 3 years. At each time point, multiple biochemical and/or physiological assessments to measure cardiovascular, respiratory, liver, renal and cognitive function, diabetes mellitus and quality of life. Recruitment was from 30 November 2017 to 31 May 2020; last follow-up measurements is due on 31 May 2023. The primary outcome measure is the incidence of new-onset type 3c diabetes mellitus during follow-up. Secondary outcome measures include: quality of life analyses (SF-36, Gastrointestinal Quality of Life Index); montreal cognitive assessment; organ system physiological performance; multiomics predictors of AP severity, detection of premature cellular senescence. In a nested cohort within the main cohort, individuals may also consent to multiparameter MRI scan, echocardiography, pulmonary function testing, cardiopulmonary exercise testing and pulse-wave analysis.

**Ethics and dissemination** This study has received the following approvals: UK IRAS Number 178615; South-east Scotland Research Ethics Committee number 16/SS/0065. Results will be made available to AP survivors, caregivers, funders and other researchers. Publications will be open-access.

**Trial registration numbers** ClinicalTrials.gov Registry (NCT03342716) and ISRCTN50581876; Pre-results.

## INTRODUCTION

Acute pancreatitis (AP) is an inflammatory disease of the pancreas usually triggered by gallstones or excess alcohol consumption.

### Strengths and limitations of this study

► Prospective cohort study with molecular mechanistic observations.
► In-depth clinical and physiological annotation of multiomics data.
► Drop-out rates in long-term prospective acute pancreatitis (AP) studies is largely unknown.
► Potential participants with severe AP may lack capacity for consent and recruitment may therefore be slower in this group.

The incidence of AP ranges from 13 to 45 per 100.00 individuals, is increasing,[1] and varies between nations. For example, USA, Finland, Scotland and Germany report higher incidences than England and The Netherlands.[2–5] AP triggers a cascade of inflammatory events causing cell damage and release of damage-associated molecular patterns that trigger inflammation and cytokine release, initiating a systemic inflammatory response resulting in multiple organ dysfunction syndrome (MODS) in one in five individuals. AP-MODS has a 21% fatality rate during the index episode, and overall in-hospital mortality is 5.2% in Scotland.[6 7] AP is graded into mild, moderate and severe, based on local and systemic determinants of severity as per the most recent AP classifications (Determinant-based Classification[8] and Revised Atlanta Classification).[9] Organ failure in AP most commonly affects the respiratory, renal and cardiovascular systems, but can involve any system.[10] The pattern of organ failure is difficult to predict and a high level of inter-individual variance is seen, regardless of the aetiology. The molecular

mechanisms that explain this inter-individual variation in local and systemic inflammatory processes in AP remain unidentified.[11] Multiomics analysis of serial time course blood samples has defined molecular patterns in human AP, called endotypes, but the precise relevance of these endotypes to clinical and therapeutic decision-making is not yet determined.[12]

Recently published data from a retrospective analysis of patients with AP has shown that the early development of MODS is associated with an increased mortality rate up to 10 years after the index presentation.[6] This strongly suggests that MODS in acute pancreatitis (AP-MODS) has a persistent and deleterious impact on patients' physiological status, though the exact nature of this pathology remains to be characterised. Existing studies indicate that most deaths are the result of cardiovascular, pulmonary, digestive or malignant disease.[13–16] In addition, the new onset of diabetes (type 3c diabetes mellitus (DM)) or impaired glucose tolerance, peripheral neuropathy and exocrine dysfunction have been documented as sources of morbidity in the years following the index attack of severe AP.[16 17] Patients with alcohol-induced pancreatitis may be at particularly increased risk of progression to chronic pancreatitis,[14] however the development of long-term complications is likely to be influenced by multiple, and as-yet unconfirmed, factors.

The aim of this observational clinical cohort study is to characterise the nature and extent of the pathophysiological impact of severe AP on organ function over the first 3 years after the index AP episode and define the long-term deleterious effect of severe AP. This will be achieved by prospectively measuring organ system function in patients recruited during a hospital admission with AP and at set time intervals thereafter. We will obtain an in-depth assessment of patients' health at presentation, and at 3 months and 36 months after the first episode of AP using markers of organ function and/or disease in the peripheral blood. In a nested cohort within the main study cohort, we will conduct cardiorespiratory evaluation tests (including exercise testing), specialised blood tests of the immune system, tests for precision medicine, and imaging to assess structure and function of key organ systems.

The results of this study will inform the design of future interventions designed to improve the long-term prognosis of patients.

## METHODS
### Ethical approval
The study will be conducted in accordance with the principles of the International Conference on Harmonisation Tripartite Guideline for Good Clinical Practice. This study has been reviewed and been given a favourable opinion by the South-East Scotland Research Ethics Committee, under the application numbers: REC Number 16/SS/0065; IRAS Number 178615 and R&D approval Lothian R&D Number 2016/0152.

### Study design
This is an observational prospective nested cohort study. The overall cohort is the population presenting with AP that meets the eligibility criteria. For the nested cohort of the study, a stratified block design is to be used to recruit equivalent numbers of patients with AP-MODS vs AP without MODS.

### Setting
Single centre academic hospital setting—Royal Infirmary of Edinburgh, NHS Lothian and the Edinburgh Clinical Research Facility, Royal Infirmary of Edinburgh, UK.

### Duration
Recruitment started on 30 November 2017 and is scheduled to end on 31 May 2020. The cohort assessment will comprise of three study visits. In-depth assessments of a participant's health at presentation, at 3 months and at 36 months follow-up study visit will be obtained. Additional cardiorespiratory evaluation tests, specialised blood tests of the immune system, tests for precision medicine, and imaging to assess structure and function of key organ systems will be conducted in a nested cohort of participants.

### Intervention
This is a prospective study examining patient outcomes and trends in patients' physiological status after an index event. Therefore, no intervention or randomisation is proposed.

### Measurements
#### Primary endpoint
► Overall cohort: to define the incidence of new-onset type 3c DM in patients with AP, compared with the age-matched population of Scotland.
► Nested cohort: to define the baseline change in pancreatic fibrosis index measured by multiparameter MRI between 3 months and 36 months after recruitment.

#### Secondary endpoints
Secondary end-points in the study will be experimental and observational, and will define explicitly, or be preliminary exploratory studies investigating:
1. Genomic predictors of AP severity and resolution of inflammation.
2. miRNA signatures of disease severity and resolution.
3. Metabolomic profiling of AP resolution, and development of local and systemic complications.
4. Premature cellular senescence as a pathological consequence of AP-MODS.
5. Alteration in immune cell subsets and phenotype as a long-term response to AP-MODS.
6. Cardiovascular and respiratory dysfunction as a legacy of endothelial injury during AP-MODS.

## Summary of clinical data to be collected

All participants in the overall and nested cohorts will have the following data collected:

1. Demographic and clinical background information.
2. Information regarding the acute presentation.
3. Organ function assessment at recruitment, 3-month and 36-month follow-up study visit:
a. Overall cohort:
   i. Full peripheral venous blood profiling, including cardiac biomarkers, standard biochemistry profiling, and samples retained for miRNA profiling, cytokines, telomere length, metabolomic profiling, proteomic profiling, transcriptomic profiling, genomic profiling, leucocyte subset analysis by flow cytometry.
   ii. Biochemical markers of organ function in urine and samples retained.
   iii. Pancreatic exocrine function test in stool (faecal elastase).
   iv. Nutritional assessment.
   v. Oral glucose tolerance test at 3-month and 36-month follow-up study visit (with the exception of insulin dependent diabetics who will have random blood glucose measured only).
   vi. 12 lead ECG, blood pressure.
   vii. Peripheral $SpO_2$.
   viii. Sway balance app non-invasive muscle function tests, 6-minute walk test.
   ix. Self-administered Patient Questionnaire
      1. Gastrointestinal Quality of Life Index.
      2. SF-12 Quality of Life.
      3. Montreal Cognitive Assessment.
b. Nested cohorts:
   i. Pulse wave analysis (with salbutamol inhaler); transthoracic echocardiography.
   ii. Spirometry and gas transfer.
   iii. Cardiopulmonary exercise (CPEX) testing.
   iv. Multiparameter MRI assessment of liver and pancreas.

### Venous blood, urine and stool sampling

Blood samples will be obtained from each study participant at each of the three study visits for organ function and specialist assays. A urine sample and a stool sample will be obtained at each of the three visits.

### Nutritional assessment

Participants will have their height and weight measured in order to calculate their body mass index, as well as bioimpedance analysis±upper arm anthropometry to assess percentage body fat and lean tissue mass. Participants will also be asked to keep a 24 hours food diary to assess whether their calorie and protein intake is sufficient for their nutritional needs at all three study visits. Trace element, including zinc, selenium and vitamin $B_{12}$ levels and other relevant micronutrients will be measured in peripheral venous blood.

### Oral glucose tolerance test

At the 3-month and 36-month follow-up study visits an oral glucose tolerance test will be performed. Participants attending the research site for physiological testing will be asked to attend in a fasted state. On arrival participants will have the baseline blood sample taken (using the same venepuncture as other tests above as appropriate), be given a standard 75 g anhydrous glucose in cold water or Polycal (113 mL) which is more palatable and will be followed by 150 mL water (total volume should be 250–300 mL). After 120 min, a second blood sample will be taken (using the same venepuncture as other tests above as appropriate) to measure blood glucose.

An oral glucose tolerance test will not be performed in participants with insulin dependent diabetes. Instead a random glucose sample will be taken at these visits.

### Non-invasive muscle function testing, Sway balance test, 6-minute walk test

Manual Muscle Testing will be performed once per visit to assess muscle strength. This will follow structured protocols used to assess muscle strength in survivors of critical illness.[18] The strength of muscles involved in shoulder abduction, elbow flexion, wrist extension, hip flexion, knee extension and ankle dorsiflexion will be graded according to the Medical Research Council muscle strength scoring system.

Participants will undergo an assessment of balance and postural stability at each visit. Testing will follow a validated protocol using the Sway Balance mobile testing system.[19 20] Participants will be asked to hold a mobile device against their chest while completing a series of stances: standing with both feet together, standing with one foot behind the other, and standing on each leg in turn.[20] Each stance is held for 10 s with the eyes closed. The Sway Balance application will generate a balance score, based on detection of alterations in position during each stance.

A 6-minute walk test will be done, according to standard protocols.[21]

### Multiparameter MRI (liver and pancreas)

A multiparameter MRI scan will be performed on each visit for the nested cohort to measure lipid, iron load and fibrosis in liver and pancreas according to protocols in development at Clinical Research Imaging Centre. The total duration of scan will be approximately 30 min.

### Cardiovascular analysis

Nested cohort participants will undergo pulse wave analysis at the radial artery using a non-invasive tonometer to measure wave reflection, pulse pressure and derived arterial stiffness with the augmentation index at 75 beats/min and the administration of two puffs of inhaled salbutamol (200 µg each puff) via a spacer device. Transthoracic echocardiography will be performed to measure longitudinal shortening of the myocardium.

## Pulmonary function testing and gas transfer

Spirometry will be undertaken in accordance with guidance from the British Thoracic Society for the nested cohort group.[22] All measurement will be made with participants seated. Forced expiratory volume in 1 second (FEV1) and forced vital capacity (FVC) will be recorded as the best of three consistent readings, and compared with the predicted normal values for the participant's height, age, gender and ethnicity. Flow volume measurements will also be obtained. Lung volumes will be measured using the steady state method.[22] Gas transfer/diffusion capacity will be measured using the single breath carbon monoxide transfer factor method.[22]

## Cardiopulmonary exercise testing (CPEX)

CPEX testing will be carried out on day 2 of patient assessment at all visits for nested cohort participants. Participants will be asked not to smoke for 8 hours prior to the test, and to abstain from other strenuous exercise on the same day.

Supervised CPEX testing will be conducted following a standardised incremental protocol on a magnetically braked cycle ergometer. Work-load ramp will be determined using a standard formula.[23] Testing will be continued until symptomatic limitation, volitional termination, failure to maintain required cadence or if other early termination conditions are met.[24] Breath by breath $O_2$ and $CO_2$ analysis will be performed using a fully calibrated metabolic cart. Twelve-lead ECG and $O_2$ saturations will be monitored continuously in the pre-test, test and recovery phase. Non-invasive blood pressure will be recorded at rest and at 3-minute intervals during test and recovery phase. The anaerobic threshold (AT) will be independently determined by two assessors using the V-slope method.

## Recruitment

All adult patients that meet the inclusion criteria at the participating sites will be recruited where possible. Patients admitted with an elevated serum amylase will be identified and members of the clinical research team will screen these patients for a diagnosis of AP using the *TRAKCare* management system (InterSystems, Massachusetts, USA). Potential participants will be approached by a member of the direct care team to confirm if they are willing to discuss the research with a member of the research team. If the individual agrees, they will be contacted by a member of the research team during their inpatient stay. If potential participants are discharged before they can be approached, they will not be included in the study

## Inclusion criteria

The following inclusion criteria will apply to all patients:
1. Confirmed clinical or radiological diagnosis of AP with an elevated serum amylase greater than 300 U/L.
2. For the potential clinical diagnosis of AP an appropriate clinical history based on compatible clinical features, will be required (ie, abdominal pain, nausea and/or vomiting), supported by the finding of elevated serum amylase greater than 300 U/L. For the radiological diagnosis, if applicable, CT and/or ultrasound scan (USS) evidence of AP will be accepted.

## Exclusion criteria

1. Patients under 16 years of age.
2. Prisoners.
3. Any patient that lacks capacity to consent.
   The additional two exclusions below apply only to those patients being considered for the nested cohort study:
4. Patients not able to undergo MRI scanning for technical reasons will be excluded (eg, those with cochlear implants, implanted pacemaker)
5. Patients with a known allergy to salbutamol
   Co-enrolment to other research studies, including drug, interventional and long-term follow-up studies, will be permitted if this has been agreed and documented by the Chief Investigators of co-enrolling studies.

## Statistical analysis plan

Full details of the statistical analysis will be documented in the statistical analysis plan developed prior to any analysis. Any deviations from this will be documented. We will present the data descriptively, where data is categorical numbers and percentages will be presented and where data is continuous number, mean, SD, median, 25th centile, 75th centile, min and max will be presented. Annual incidence of type 3c diabetes will be given as a percentage with an accompanying 95% CI. For the nested cohort we will present the change in fibrosis index over time graphically with accompanying descriptive statistics, and where appropriate include details of the index episode including aetiology, severity and pre-existing medical background as covariates wherever possible. Quality of assessment will be included as a potential confounder.

## Sample size calculation

Overall sample size calculation has been performed to illustrate how well we will be able to estimate the annual incidence of type 3c diabetes in patients with AP with varying sample sizes.[25 26] The incidence of type 3c DM was chosen for the purposes of establishing a prospective cohort size because it is specific, measurable, clinically relevant and offers a practice-changing opportunity for intervention. With a sample of 323 we would have 80% power to be able to show a difference from the incidence of DM in the general population (6%) if we observed an incidence in the cohort of 10% or greater. Moreover, this sample size will support the evaluation of the secondary endpoints, allowing us to adjust for any confounding factors.

Nested cohort sample size has been calculated to detect the difference in the 3 months to 36 months change in pancreatic fibrosis index between those with AP with

MODS and those with AP without MODS based on Banerjee *et al* estimates of fibrosis index using multiparameter MRI.[27] Based on a sample size of 20 participants per group, using a two-sided two-sample test with a 5% level of significance and 80% power, we would be able to detect an effect size of 0.909 (where the effect size is the absolute difference in means divided by common SD). To allow for potential dropouts in this part of the study (death, withdrawal) a minimum sample size of 25 individuals will be recruited for the nested cohort.

### Data management

Data will be collected either directly onto an electronic case report form on Research Electronic Data Capture tools hosted at University of Edinburgh[28 29] or onto paper data collection sheets which will be entered onto the electronic form at site. All participants will be allocated a unique study number which will be used in all electronic and paper data collection tools. The study link-anonymised data will be stored on secure password-protected, regularly backed-up, computerised storage in accordance to secure data protection principles of the University of Edinburgh/clinical research facility (CRF) at study sites.

Personal contact details and the Community Health Index (CHI) number are required for follow-up and tracking electronic health records, and will initially be kept on paper at participating sites in a secure storage area with restricted access. These details will ultimately be stored on an encrypted database within NHS Lothian. Personal contact details and CHI numbers will not be entered onto the study database.

The chief investigator (DJM) will have access to the data during all stages of the study. The analysis will be performed in secure premises at the University of Edinburgh and NHS Lothian by the chief investigator and the assigned statistician. The chief investigator will have control of and act as the custodian for the data generated by the study.

### Participants

#### Consent

Potential study participants will be provided with a written information sheet and a written consent form. In addition, participants will meet with a member of the CRF team to receive a full explanation of the nature and purpose of the study, with the opportunity to ask questions. It will be made clear that the individual may withdraw from the study at any time without giving a reason and with no consequence to their current or future care. Recruitment and informed consent will be performed by the investigator/member of the CRF team at the participating site. If individuals wish to participate, they will sign a consent form, countersigned by the investigator/CRF researcher. In addition, consent will be obtained to inform the participants' GP and Consultant in charge of their care.

#### Patient decision period

Potential participants who have been invited to participate will be given a period of up to 2 days (minimum 1 hour) to read the relevant information sheets, to contact a member of the study team to get further information and ask questions, before they are invited to participate in a conversation and commence the process of fully-informed consent.

### Withdrawal of participants

Participants will be withdrawn from the clinical research study in the following circumstances:
1. In cases of withdrawal of informed consent, where capacity exists.
2. In cases of withdrawal of consent by the participant's representative for adults with incapacity.

Participants who wish to withdraw from the study will be given the option to permit ongoing use of data and samples that have already been collected, and/or future recording and usage of routinely collected clinical data and results. This will be clearly documented on the patient consent form. Patients that become uncontactable will have any previously collected data included in the study unless consent is withdrawn as above.

### Benefits to participants

No direct benefit is expected for the participants. We envisage that the results of this study will benefit other patients with AP in the future, and society as a whole by advancing scientific knowledge on this condition. We believe that this study may lead to new strategies in the treatment of AP which will prevent or ameliorate the impact of organ dysfunction. This will help to maximise the health, and life expectancy, of patients with AP.

### Potential burden to participants

The participants will be subjected to peripheral venous blood sampling which may cause discomfort or pain. In order to minimise these effects, blood sampling will be performed by appropriately trained and experienced members of the research team, in adherence to the local guidelines for blood sampling. Where possible, and only if the specific study venepuncture time-points coincide, clinical care samples will be taken at the same time to reduce the number of venepunctures that any one participant sustains.

The collection of urine and stool samples is not expected to cause any harm or discomfort; however, some minimal inconvenience may be caused. In order to address this, a convenient time for the participants and the clinical care team will be chosen, and a full explanation of the purpose of these tests will be provided. The participants' dignity and privacy will be respected at all times.

The 6-minute walk test is not anticipated to cause any harm or discomfort to participants. Pulse wave analysis is not expected to cause any significant adverse effects. Inhaled salbutamol may cause a very slight peripheral tremor, but no other expected harm or discomfort.

CPEX testing is associated with a very low risk of serious complications; however, this is related to the severity of co-morbid disease such as coronary artery disease (CAD).[24] For those participants with identified CAD, the opinion of a cardiologist will be sought on the suitability of the participant for this test.

Assessment of pulmonary function using spirometry is not known to be associated with any significant adverse events.[30] Participants may experience some discomfort or dizziness during the test due to the requirement for complete exhalation.[31] Spirometry is associated with increased myocardial strain, and a rise in intrathoracic, intraabdominal and intraocular pressures, and consequently will not be performed on participants with the following: recent head/neck, thoracic or abdominal surgery, recent myocardial infarction, retinal detachment, resting pulse greater than 120 beats/min, pregnant, severe heart failure or arrhythmia, pneumothorax, aortic or cerebral aneurysm.[32]

MRI scans are not expected to pose risks to the majority of participants, in the absence of known contraindications. Participants will be assessed prior to the scan for any potential contraindications to the procedure such as mechanical heart valves, pacemakers, metallic prostheses and cochlear implants.[33]

## ETHICS AND DISSEMINATION

This study has been reviewed as IRAS Number 178615 by the South-East Scotland Research Ethics Committee under the application numbers (REC Number) 16/SS/0065 and has been given a favourable opinion. NHS Scotland R&D approval have been granted. The study sponsor is NHS Lothian/University of Edinburgh ACCORD.

Results of the study will be presented at local, national and international medical meetings. The findings of the study will be published in peer-reviewed medical/scientific journals and made open-access on acceptance. If appropriate, the results of the study will be disseminated via press releases by the University of Edinburgh. Information may also be disseminated to the general public via public engagement and community outreach programmes.

### Patient and public involvement

Patients who have had an episode of AP and who participate in the acute pancreatitis patient liaison group were invited to input and contributed to the concepts and the trial design.

### Versioning

This paper is based on the full protocol V.11, dated 23 September 2019

#### Author affiliations
[1]Clinical Surgery, University of Edinburgh, Edinburgh, UK
[2]Edinburgh Clinical Trials Unit, University of Edinburgh, Edinburgh, UK
[3]Clinical Research Facility, NHS Lothian, Edinburgh, UK
[4]Centre for Inflammation Research, University of Edinburgh, Edinburgh, UK
[5]Anaesthesia and Critical Care, NHS Lothian, Edinburgh, UK
[6]Respiratory Physiology, NHS Lothian, Edinburgh, UK
[7]Centre for Cardiovascular Science, University of Edinburgh, Edinburgh, UK
[8]Epidemiology and Statistics Core, Edinburgh Clinical Research Facility, University of Edinburgh, Edinburgh, UK

**Contributors** DJM had the idea and wrote the funding application with statistics input from CG. DJM, CV, CG and JB developed and drafted the clinical study

protocol and ethics application. XZ, AuW, DCC, SB, JM, SS and DM designed and contributed additional details of the investigations. MS and AnW designed and built the data collection system. DJM, JB, MG, KW, AW, LM, EC, ET, AES, DD, RM, SC-S and other RESORP team members initiated the project. AES, RM, DJM drafted the manuscript.JM devised the lung function portion of the protocol and SB devised the Cardiopulmonary Exercise test (CPEX) protocol for the study. All authors contributed to and approved the final version of the manuscript.

**Funding** This study is funded by the United Kingdom Medical Research Council through a Senior Clinical Fellowship to Damian J. Mole. Ref: MR/P008887/1.

**Disclaimer** The sponsor and funder has no direct role or influence in the design or conduct of the study, or the decision to submit the protocol for publication.

**Competing interests** DJM wishes to declare that he holds a senior position in a private company developing medicines for systemic inflammation and cancer, and that he has previously received funding for collaborative studies on AP from GSK.

**Patient and public involvement** Patients and/or the public were involved in the design, or conduct, or reporting, or dissemination plans of this research. Refer to the Ethics and Dissemination section for further details.

**Patient consent for publication** Not required.

**Provenance and peer review** Not commissioned; externally peer reviewed.

**ORCID iD**
Damian J Mole http://orcid.org/0000-0001-6884-7302

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
