## [Reviewer comments · BMJ Open]

ARTICLE DETAILS

TITLE (PROVISIONAL)	Study Protocol for RESORP – Resolution of Organ Injury in Acute Pancreatitis – an observational prospective cohort study.
AUTHORS	Sherif, Ahmed Elshawadfy; McFadyen, Rory; Boyd, Julia; Ventre, Chiara; Glenwright, Margaret; Walker, Kim; Zheng, Xiaozhong; White, Audrey; McFadyen, Laura; Connon, Emma; Damaskos, Dimitrios; Steven, Michelle; Wackett, Anthony; Thomson, Euan; Cameron, David C; Semple, Scott; Morris, David; Clark-Stewart, Saskia; Graham, Catriona; Mole, Damian J.

VERSION 1 – REVIEW

REVIEWER	PRAMOD GARG A.I.I.M.S., New Delhi.
REVIEW RETURNED	17-Jun-2020

GENERAL COMMENTS	The “Study protocol for RESORP- Resolution of Organ Failure in Acute Pancreatitis” submitted by Sherif et al is an important step to assess long-term consequences of acute pancreatitis. Following comments might be useful:  1. I am not sure if pancreatitis fibrosis is a good co-primary outcome unless one correlates it with extent of pancreatic necrosis at the index admission and recurrent pancreatitis during follow up, and as a biomarker for early CP. 2. Micro and macro nutrient deficiencies due to poor diet and/or maldigestion may develop which may contribute to overall health and possibly to organ dysfunction since acute pancreatitis is an illness which takes months to recover and patients can't revert back to their normal diet due to various reasons. 3. Sarcopenia could be another important variable that can be assessed by MRI done at 3 months and 3 years. 4. The aetiology of AP such as alcohol may also influence organ dysfunction and should be adjusted for. 5. The quality of assessment is another important outcome which could be added as an outcome.
--

REVIEWER	Kaijun Liu Department of Gastroenterology, Daping Hospital, Army Medical University, Chongqing, China.
REVIEW RETURNED	27-Jun-2020

GENERAL COMMENTS	This study protocol is aiming to characterise the precise nature and extent of organ dysfunction following AP. This study is well designed, but some questions should be addressed before acceptance.  1. Line 36, what does "an acute university hospital" mean? 2. Line 148, The aim of this study is to investigate the characteristics
---

	of organ dysfunction following AP. However, the primary endpoint only look into type 3c DM. Why do the authors only make type 3c DM as the primary endpoint? 3. Line 153, the secondary endpoints included genomic predictors, miRNA signitures, metabolic profiling, et al. After recuitment, AP has been confirmed and the severity of AP has also be defined. Why do investigators still want to detect the gemomic predictor of AP severity, miRNA signatures of disease severity? Moreover, could authors give more detail of the predictors, miRNA signitures, metabolic profiling et al? 4. Line 276, could authors provide more details on the inclusion criteria? for example, age, severity of AP, et al. 5. Line 300, authors only descriped how to present the data. Could authors provide more details on the statistics of the study? For example, how to analyze the data of the three time points of the cohort? 6. Line 233, could authors provide more details of the MRI sequences which would be used in the study?
--	--

VERSION 1 – AUTHOR RESPONSE

Reviewer 1:

Thank you for your time and insight into our work and your helpful suggestions.

1. I am not sure if pancreatitis fibrosis is a good co-primary outcome unless one correlates it with extent of pancreatic necrosis at the index admission and recurrent pancreatitis during follow up, and as a biomarker for early CP.

This is an interesting point and this was already the intention, but thank you for requesting that it is made clearer in the text. Certainly the extent of pancreatic necrosis will be factored in to the baseline estimation of the severity of the index episode. However, the specific nested cohort primary outcome is the change between 3 and 36 months for a given individual, so each individual is also their own control. The point is valid and where appropriate we will include the severity/extent of local pancreatic injury during the index episode as a co-variate. Text has been added to the statistical analysis plan to this effect (lines 310 to 312).

2. Micro and macro nutrient deficiencies due to poor diet and/or maldigestion may develop which may contribute to overall health and possibly to organ dysfunction since acute pancreatitis is an illness which takes months to recover and patients can't revert back to their normal diet due to various reasons.

This is correct. These variables are being measured as follows:

“Participants will have their height and weight measured in order to calculate their Body Mass Index (BMI), as well as bioimpedance analysis +/- upper arm anthropometry to assess percentage body fat and lean tissue mass. Participants will also be asked to keep a 24-hour food diary to assess whether their calorie and protein intake is sufficient for their nutritional needs at all 3 study visits.” We have added the following text:

“Trace element, including zinc, selenium and B12 levels and other relevant micronutrients will be measured in peripheral venous blood.” (lines 208 to 209)

3. Sarcopenia could be another important variable that can be assessed by MRI done at 3 months and 3 years.

This is an interesting and helpful suggestion. We will endeavour to do this if those data can be extracted from the exisiting sequences, but are not able to change the MRI acquisition parameters at this stage, and have therefore not changed the protocol.

4. The aetiology of AP such as alcohol may also influence organ dysfunction and should be adjusted for.

We agree, and this has always been the intention. We have highlighted the intention to include this covariate in the analysis (lines 310 to 312).

5. The quality of assessment is another important outcome which could be added as an outcome. Thank you for this comment; we regard this to be more of a confounding variable rather than an outcome per se, but will certainly factor this in to the analysis. We have added text to the statistical analysis section to reflect this (lines 310 to 312).

Reviewer 2:

Thank you for your time and insight into our work, and your interesting questions:

1. Line 36, what does "an acute university hospital" mean?

An acute University hospital means a University hospital with a receiving unit for acute and emergency admissions (as opposed to an elective-only centre, for example a cancer centre). We have altered the sentence to improve the clarity (line 37).

2. Line 148, The aim of this study is to investigate the characteristics of organ dysfunction following AP. However, the primary endpoint only look into type 3c DM. Why do the authors only make type 3c DM as the primary endpoint?

In a cohort study such as this, for the purposes of establishing a prospective cohort size for statistical power calculations, we believe it is good practice to select one primary end-point on which to base that. It should be specific, measurable, clinically relevant and ideally offer a practice changing opportunity for intervention. Therefore, type 3c DM was chosen as it fulfilled these characteristics. Multiple additional end-points are being recorded and will be reported, as described. Text to this effect has been inserted in the sample size calculation section (lines 317 to 319).

3. Line 153, the secondary endpoints included genomic predictors, miRNA signatures, metabolic profiling, et al. After recruitment, AP has been confirmed and the severity of AP has also be defined. Why do investigators still want to detect the genomic predictor of AP severity, miRNA signatures of disease severity? Moreover, could authors give more detail of the predictors, miRNA signatures, metabolic profiling et al?

We include these analyses in order to identifying potential biomarkers, diagnostics and potentially tractable mechanisms of disease for future patients. The analysis will be exploratory using open multiomics approaches, as described.

4. Line 276, could authors provide more details on the inclusion criteria? for example, age, severity of AP, et al.

As already specified in line 269, all adult patients will be included if they meet the specified inclusion criteria (lines 279 to 287), and if they do not meet the exclusion criteria as specified (lines 288 to 296).

5. Line 300, authors only described how to present the data. Could authors provide more details on the statistics of the study? For example, how to analyze the data of the three time points of the cohort?

As stated, the statistical analysis plan will be developed prior to any analysis and deviation documented. No further detail is intended to be supplied at this point.

6. Line 233, could authors provide more details of the MRI sequences which would be used in the study?

The MRI protocols have been developed for this study and will be reported separately as part of the study results.

We hope that you and the reviewers find these revisions acceptable and that our revised protocol paper is suitable for publication in BMJ Open. Thank you again for taking the time to consider our work.

VERSION 2 – REVIEW

REVIEWER	Pramod Garg All India Institute of Medical Sciences, New Delhi
REVIEW RETURNED	07-Aug-2020

GENERAL COMMENTS	Thank you. It is a well planned study.
--

REVIEWER	Kaijun Liu Department of gastroenterology, Daping Hospital, Army Medical University, Chongqing, China.
REVIEW RETURNED	10-Aug-2020

GENERAL COMMENTS	Authors answered part of my questions, and there is no "response to reviewer letter ". Please answer questions point to point, even if you disagree with my view or my question.
--